# The Effect of Pregnancy and Inflammatory Bowel Disease on the Pharmacokinetics of Drugs Related to Inflammatory Bowel Disease—A Systematic Literature Review

**DOI:** 10.3390/pharmaceutics14061241

**Published:** 2022-06-11

**Authors:** Thomas K. Wiersma, Marijn C. Visschedijk, Nanne K. de Boer, Marjolijn N. Lub-de Hooge, Jelmer R. Prins, Daan J. Touw, Paola Mian

**Affiliations:** 1Department of Clinical Pharmacy and Pharmacology, University Medical Center Groningen, 9713 GZ Groningen, The Netherlands; t.k.wiersma@student.rug.nl (T.K.W.); m.n.de.hooge@umcg.nl (M.N.L.-d.H.); d.j.touw@umcg.nl (D.J.T.); 2Department of Gastroenterology and Hepatology, University Medical Center Groningen, 9713 GZ Groningen, The Netherlands; m.c.visschedijk@umcg.nl; 3Department of Gastroenterology and Hepatology, AGEM Research Institute, Amsterdam University Medical Centre, Vrije Universiteit Amsterdam, 1081 HV Amsterdam, The Netherlands; khn.deboer@amsterdamumc.nl; 4Department of Obstetrics and Gynecology, University Medical Center Groningen, 9713 GZ Groningen, The Netherlands; j.r.prins@umcg.nl

**Keywords:** inflammatory bowel disease, pregnancy, pharmacokinetics

## Abstract

Due to ethical and practical reasons, a knowledge gap exists on the pharmacokinetics (PK) of inflammatory bowel disease (IBD)-related drugs in pregnant women with IBD. Before evidence-based dosing can be proposed, insight into the PK has to be gained to optimize drug therapy for both mother and fetus. This systematic review aimed to describe the effect of pregnancy and IBD on the PK of drugs used for IBD. One aminosalicylate study, two thiopurine studies and twelve studies with biologicals were included. Most drugs within these groups presented data over multiple moments before, during and after pregnancy, except for mesalazine, ustekinumab and golimumab. The studies for mesalazine, ustekinumab and golimumab did not provide enough data to demonstrate an effect of pregnancy on concentration and PK parameters. Therefore, no evidence-based dosing advice was given. The 6-thioguanine nucleotide levels decreased during pregnancy to 61% compared to pre-pregnancy levels. The potentially toxic metabolite 6-methylmercaptopurine (6-MMP) increased to maximal 209% of the pre-pregnancy levels. Although the PK of the thiopurines changed throughout pregnancy, no evidence-based dosing advice was provided. One study suggested that caution should be exercised when the thiopurine dose is adjusted, due to shunting 6-MMP levels. For the biologicals, infliximab levels increased, adalimumab stayed relatively stable and vedolizumab levels tended to decrease during pregnancy. Although the PK of the biologicals changed throughout pregnancy, no evidence-based dosing advice for biologicals was provided. Other drugs retrieved from the literature search were mesalazine, ustekinumab and golimumab. We conclude that limited studies have been performed on PK parameters during pregnancy for drugs used in IBD. Therefore, more extensive research to determine the values of PK parameters is warranted. After gathering the PK data, evidence-based dosing regimens can be developed.

## 1. Introduction

Inflammatory bowel disease (IBD) is an overarching term for chronic inflammation in the gastrointestinal tract [1]. IBD is characterized by exacerbations. Medications play a main role in maintaining the remission of IBD. Considering the main drug classes used as therapy, aminosalicylates, thiopurines, corticosteroids, immunosuppressants, biologicals and JAK-inhibitors play a dominant role [2]. The two most common variants of IBD are ulcerative colitis (UC) and Crohn’s disease (CD). A Dutch population-based cohort study found that, among 2837 IBD patients, 59% had UC and 41% had CD [3]. The exact mechanism of developing IBD is unknown. However, there is consensus about the multifactorial characteristics of its onset. Globally speaking, genetic factors, environment, immune response and intestinal barriers are the most important factors for the development of IBD. If those factors change, they will exert influence over the microbiome in the intestines, potentially leading to IBD [4].

Because the diagnosis of IBD is frequently made in the fertile period of women, pregnancy often coincides with IBD [5]. A consensus exists among clinicians to resume the treatment of pregnant women with IBD. The behavior of the drug is evaluated for its efficacy and toxicity, taking into account both the mother and fetus. It is of utmost importance to maintain IBD in remission to avoid adverse pregnancy outcomes, such as miscarriages and pre-term birth. Of major importance in continuing therapy is the dosing of drugs [5]. It has to be noted that pregnancy is associated with physiological changes (e.g., increased body water, changed metabolic enzyme expression and renal function) that influence the pharmacokinetics (PK) of many drugs. Based on the physiological changes taking place during pregnancy, PK is often different in pregnant women, and dosage adaptations are necessary [6]. Currently, pregnant women are often administered the same dose as non-pregnant women. However, PK changes may lead to either subtherapeutic or toxic drug concentrations in mother and/or fetus. Furthermore, irrespective of whether the fetus is a target of pharmacotherapy, it is probably exposed to any drug taken by the mother [7]. Although questions concerning effective dosing of individuals often arise in clinical settings, dosing in pregnant women is still empirical instead of evidence-based, as pregnant women are excluded from clinical trials. To develop evidence-based dosing in pregnant women, insight has to be gained in PK of drugs.

Therefore, this study aimed to systematically describe the effect of pregnancy and IBD on the PK of drugs used in IBD therapy and to investigate if, based on the possible changed PK, evidence-based dosing guidelines can be developed.

## 2. Materials and Methods

### 2.1. Search Strategy

This systematic review of the literature was performed in accordance with the PRISMA guidelines of 2020 [8]. A systematic search was conducted by using PubMed on 10th January 2022 to retrieve studies on the PK of IBD-related drugs throughout different trimesters of pregnancy or in women at delivery. English or Dutch written articles, without limit to publication date, were included. The search strategy consisted of three main keywords” “pharmacokinetics”, “IBD-related drugs” and “pregnant women”. For the specific keywords and field codes per topic, see Table 1. The IBD-related medication key terms with accompanying field codes are provided in Appendix A Table A1. In addition, the references of the included studies were checked for relevant articles.

### 2.2. Study Selection

First the title and abstract were screened for the three main topics (Table 1). When a study included all three topics, the full article was studied. In a later phase, a distinction between IBD patients and non-IBD patients was made. Studies including non-IBD participants were excluded. Studies not meeting the study aim and inclusion criteria were excluded. Two investigators (TW and PM), separately from each other, conducted the search strategy and the study selection. The obtained results were discussed, and in the case of disagreement, a third author (DT) was consulted.

### 2.3. Data Extraction

When the studies were included in this study, the data were extracted in a Microsoft Excel sheet. The data extraction was performed separately by two investigators (TW and PM) for all included studies. In the case of disagreement, a third author (DT) was consulted. The study characteristics of interest that were extracted were the study design, number of pregnant women included in the study, type of medication (with dosage and dosing interval), moment in time when participants were studied (before pregnancy (T0), trimester 1 (T1), 2 (T2), 3 (T3), at delivery (T4) and/or postpartum (T5)), age and bodyweight at inclusion, type of IBD and the analytical method used for drug concentration measurements. The timeframes for the trimesters were defined as 0 to 13 weeks for T1, 14 to 26 weeks for T2 and 27 to 40 weeks for T3. In addition, the PK parameters per study were extracted. Furthermore, it was investigated if, based on potentially changed PK during pregnancy, adapted dosages were advised by the studies. When the numerical values for the PK parameters were not available within the study, but a graph was available, the data were extracted by using R version 4.1.2 and R Foundation for Statistical Computing, Vienna, Austria, with the use of the package Digitize version 0.0.4.

## 3. Results

### 3.1. Study Selection

A total of 430 studies were identified. After the removal of duplicates (*n* = 36) and removal of 341 studies based onnot meeting the criteria, full texts were obtained from 53 studies, of which 37 were excluded. The reasons for exclusion are provided in Figure 1. These include, among other reasons, ex vivo data, non-applicable outcomes for this study, data related only to the fetus or infant, or letters to the editor as a reaction on publications. Consequently, 15 studies were included. One study covered aminosalicylates, two studies thiopurine therapy and 12 studies biologicals. The PRISMA flow diagram is presented in Figure 1.

### 3.2. Aminosalicylates

One aminosalicylate study was found [9]. In this study, the outcomes of five participants were found to be suitable for this review. One woman used a suppository, three women used a tablet and one woman used a combination of both drugs. The age and weight of these women were not provided. The drug concentration was measured at delivery, with a timeframe from dosing to delivery ranging from 5 to 24 h. The lowest concentration was found in a patient using only the mesalazine tablet, with a concentration of 0.2 μmol/L. The highest concentration was found in another patient using only the tablet, with a concentration of 2.6 μmol/L.

In conclusion, based on the limited existing data, no conclusion can be drawn on possible changes in the PK of aminosalicylates throughout pregnancy. Furthermore, based on the limited available data, no evidence-based dosing regimen could be provided.

### 3.3. Thiopurines

Two studies focused on the pro-drug azathioprine (AZA) and mercaptopurine (MP) [10,11]. AZA is converted mainly by glutathione S-transferases into MP. A big portion of MP is then metabolized by thiopurine-S-methyltransferase into the metabolite 6-methylmercaptopurine (6-MMP). Another portion of the MP will be metabolized via the purine salvage pathway into the three nucleotides, 6-thioguanine monophosphate, 6-thioguanine diphosphate and 6-thioguanine triphosphate. The enveloping name of these three nucleotides is 6-thioguanine nucleotides (6-TGN). Since 6-MMP and 6-TGN are the metabolites of interest for the therapy, the studies reported their outcomes in the levels of these metabolites [12]. When those two studies were combined, the total amount of participants included was 72. The percentage of participants with UC was 25%, CD 71% and undetermined IBD 4%. The patient and study characteristics are elaborated in Appendix B Table A2. The participants using AZA (71%) were more prevalent than MP (29%). The studies showed similarities on multiple aspects in their analytical quantification methods. Quantification occurred by using a modified Dervieux method. Both measured the active metabolites of AZA and MP, namely 6-TGN and its potentially toxic variant 6-MMP in red blood cells (RBCs). The results are presented in Table 2. All measurements were performed from pre-pregnancy until after the delivery. 

Both studies show the same phenomenon when studying the changes of 6-TGN and 6-MMP levels throughout pregnancy. The 6-TGN levels are lower during the first, second and third trimesters compared to preconception. The most noticeable differences per trimester compared to pre-pregnancy levels were found in the study by Flanagan et al. and are as follows: T1 with 83%, T2 with 61% and T3 with 73% [10]. In contrast, the 6-MMP levels increased during pregnancy compared with the preconception state. The most extensive alteration per trimester was shown by Jharap et al., with 166% in T1 [11]; Flanagan et al., with 209% in T2 [10]; and Jharap et al. in T3, with 205% [11] compared to pre-pregnancy levels. After delivery, both 6-TGN and 6-MMP levels returned to the preconception baseline levels. Figure 2 and Figure 3 show the differences in metabolite levels for both studies over time. Although the PK of the thiopurines changed throughout pregnancy, no evidence-based dosing advice was provided. One study [10] suggested that caution should be exercised in case the doses are to be changed during pregnancy. This advice is based on their observation of shunting 6-MMP levels due to dosage change. An increase in thiopurine dose is sometimes inevitable, as a consequence of rising 6-MMP levels. However, if the 6-MMP levels stay below the toxic threshold and toxic side effects are absent, alterations in dosage seems to be possible.

In conclusion, two studies were available that covered the PK of thiopurines during pregnancy [10,11]. The therapeutic 6-TGN levels decreased during pregnancy, while the potential toxic 6-MMP levels rose. Although the PK of the thiopurines changed throughout pregnancy, no evidence-based dosing advice was provided. One study [10] advised to be cautious when the dosage is altered by monitoring for toxic side effects and keeping the 6-MMP levels below the toxic threshold.

### 3.4. Biologicals

A total of 12 studies [13,14,15,16,17,18,19,20,21,22,23,24] were included, of which four studies [14,15,17,21] presented data on more than one drug. Five unique drug types were found, namely infliximab (IFX), adalimumab (ADL), vedolizumab (VDZ), ustekinumab (UST) and golimumab (GLM). Respectively, nine, four, two, two and one articles provided data on these drugs. The cumulative number of enrolled participants was 173. The number of CD, UC and IBD unspecified women were 112 (70%), 46 (29%) and 2 (1%), respectively. The article of Bortlik et al. [20] was excluded from the previous sum, because the authors did not provide a distinction in CD and UC, and it was unspecified from the total IBD.

#### 3.4.1. TNF-α Inhibitors—Infliximab, Adalimumab and Golimumab

Within the group of the TNF-α inhibitors, nine studies presented suitable data for IFX, four studies for ADL and one study for GLM. The cumulative numbers of observed participants per drug were 83, 40 and 1 for IFX, ADL and GLM, respectively. When all participants within this group were combined (excluding the study of Borlik et al.), 75% were diagnosed with CD, 23% with UC and 2% with unspecified IBD. The range of median and mean ages was between 28.9 and 36 years within the studies (Appendix B Table A2). In the case of IFX and ADL, respectively, four and two studies presented data over multiple timeframes. The study that covered GLM only presented data at delivery. Five studies measured IFX data at one point, being three studies at delivery and two studies after pregnancy. In the case of ADL, two studies obtained their data at delivery. The included TNF-α inhibitor studies were predominantly found to be prospective cohort studies, four covering ADL and six IFX. One study was found to be a retrospective cohort study obtaining data from participants using IFX. Lastly, two IFX studies and one GLM study were determined as a case report (Appendix B Table A2).

When focusing on the PK parameters, all studies within the group of TNF-α inhibitors reported either the trough concentration (Ctrough), *n* = 4, or unspecified concentration (Cunspecified), *n* = 6 (Table 2). One study used a population PK model to determine the clearance and volume of distribution for IFX [21]. They reported a clearance of 0.608 L/d and volume of distribution of 18.2 L. Four and two studies presented data on multiple time points throughout different stages of pregnancy for IFX and ADL, respectively. The data of these studies are shown in Figure 4 and Figure 5, respectively.

For IFX (Figure 4), the authors, who measured IFX levels pre-pregnancy, during pregnancy and postpartum, determined an increase during pregnancy compared to pre-pregnancy levels [14,15,21]. The highest percentage of increase compared to pre-pregnancy levels was 123% [14] in the first trimester, 205% [21] in the second trimester and 305% [14] in the third trimester. The IFX levels after delivery compared to pre-pregnancy levels were higher in Seow et al. [14] and Flanagan et al. [15] (10.17 against 6.9 μg/mL and 10.3 against 7.9 μg/mL, respectively) and were lower in the study of Grišić et al. [21] (5.9 against 7.3 μg/mL). The IFX levels after pregnancy were all lower than during pregnancy. However, the degree in change was different among studies. Lastly, Figure 4 shows a large dispersion in data at the after-delivery moment. Two studies seem to have high concentrations compared to all other studies [13,18]. It has to be noted that the high variability (Figure 4) is probably due to the fact that these studies were case reports in which outliers are more easily visible compared to a median or mean values represented in cohort studies.

In Figure 5, the ADL concentrations over time are provided. The authors [14,15] mentioned that the ADL concentration during pregnancy is relatively stable compared to the ADL concentration before and after pregnancy. It is, however, observed that pregnancy levels are lower than pre-pregnancy levels. Furthermore, the second trimester contains a discrepancy. No reason for this discrepancy was reported by the authors.

Not all included studies provided dose advice, and when they did, it was general advice [14,15,17,19,20,21]. However, one study from Steenholdt et al. [17] provided a specific target advice. A concentration of 0.5 mg/mL was considered as a therapeutic threshold [17]. Looking at the other studies presenting dose advice for IFX, the following results were found. Four studies mentioned that dosing for IFX should be halted at the end of the second trimester or the beginning of the third trimester [14,19,20,21]. The main reason for above-mentioned advice is to suppress, as much as possible, immune response after birth.One study suggested that the dose could be changed during pregnancy to the lower end of the therapeutic range [14], while another study did not recommend a change in dose [15]. Although the PK of biologicals changed throughout pregnancy, none of the studies indicated how dosing should be adapted during pregnancy. The same is applicable to ADL, where three studies [14,19,20] advised to stop dosing after the second trimester. No specific dosing advice was provided for earlier trimesters based on changed PK data. The same study as with IFX saw no problem in changing the dose during pregnancy to the lower end of the therapeutic range, while another study did not recommend a dose change [14,15]. For GLM, no dose advice was given by the authors.

#### 3.4.2. Integrin Inhibitor—Vedolizumab

Two studies provided data for a total of 28 pregnant women with IBD. Of these women, 42% were diagnosed with CD and 58% with UC. The median age of the participants was 30.7 years in the study of Flanagan et al. and 31 years in the study of Mitrova et al. [15,24]. The study of Flanagan et al. [15] provided data over multiple moments within the pregnancy-until-delivery timeframe, namely T1 19.1 (13–23), T2 15.1 (8.6–21.7), T3 9.5 (3.7–20.0) and T4 5.5 (1.1–9.9) μg/mL. The study of Mitrova et al. [24] solely presented data at delivery, namely 7.3 (2.9–17.9) μg/mL. Both studies were prospective cohort studies.

Concerning the PK data, both studies presented their values as concentrations. Flanagan et al. presented their concentration as trough levels, while Mitrova et al. did not specify their type of concentration. The data of both studies are presented in Figure 6.

Flanagan et al. mentioned that no dose change was recommended for VDZ [15].

#### 3.4.3. Interleukin 12/23 Inhibitor—Ustekinumab

Two studies focused on the use of UST in pregnant women with IBD. In total, 16 participants were included, of which 94% were diagnosed with CD and 6% with UC. The study of Mitrova et al. [24] was a prospective cohort study in which the median age was 28 years. The study of Sako et al. was a case report in which the woman was 35 years of age [22]. Both studies presented their data as unspecified concentration, only at delivery. Since the concentration was only available at delivery, it was not possible to see the behavior of the UST concentration during the pregnancy. As a consequence, due to a lack of data, the authors of these articles could not provide a dose advice.

In conclusion, 12 studies were found that presented drug concentrations for IFX, ADL, VDZ, UST and GLM. Most studies [16,19,20,22,23,24] only presented a concentration at delivery. The studies that presented data during the whole pregnancy showed an increase in concentration for IFX, a relative stable concentration for ADL and a decreasing concentration for VDZ. Although the PK of the biologicals changed throughout pregnancy, no evidence-based dosing advice was provided. One study presented a target advice, being that 0.5 mg/mL for IFX seemed to be a therapeutic concentration.

## 4. Discussion

To our knowledge, this is the first time a systematic review was conducted on the available data of PK parameters related to IBD drugs in pregnant women with IBD. Our ultimate goal was to provide an in-depth overview of the available PK data. Before evidence-based dosing can be proposed for pregnant women, insight into the PK has to be gained to optimize drug therapy for both the mother and fetus. Limited PK studies on IBD drugs have been performed during pregnancy, and, in general, they have not resulted in obtaining PK parameters in the different pregnancy trimesters. Although the PK of the IBD-related drugs changed throughout pregnancy, no evidence-based dosing advice was provided.

When focusing on the present guidelines of the European Crohn’s and Colitis Organization (ECCO) and the American Gastroenterological Association (AGA), both state that staying in remission is important to minimize adverse outcomes. Therefore, non-teratogenic medication should be used during pregnancy in order to reduce the chance of a flare during pregnancy [25,26]. Flares are a risk factor for adverse outcomes for both the fetus (e.g., increased chance of preterm birth and low birth weight) and for the mother (e.g., emergency caesarian delivery and thromboembolic events) [25,27]. The ECCO and AGA consider mesalazine, sulfasalazine, thiopurines, biologicals and corticosteroids (for a short period) to be safe when used for maintenance therapy during pregnancy. Tofacitinib is a relatively new small-molecule drug with limited human data in pregnancy. The producer and the AGA suggest that tofacitinib should not be advised to be used, especially not in the first trimester of pregnancy [26,27].

Our systematic review shows that concentrations of IBD drugs vary during the different trimesters of pregnancy. However, since information is too lacking to give dosing advice, there is a need to expand the study duration over multiple trimesters to obtain the PK of IBD drugs. In this discussion, the most important findings arising from this review and the still remaining PK-related knowledge gaps are discussed. Furthermore, a recommendation is made regarding information that still needs to be collected in order to develop evidence-based dosing for IBD-drugs in pregnant women with IBD, while also taking the fetus into account.

One study was found in which concentrations at delivery were presented for mesalazine [9]. No dose advice and different dosages and routes of administration, in combination with only five participants studied, made us question the usability of this study. We conclude that this commonly used drug in IBD is overlooked in the literature. Only two prospective studies presented concentrations of the thiopurine metabolites 6-TGN and 6-MMP. Both studies found that, during pregnancy, the therapeutic 6-TGN levels decreased, while the potential toxic 6-MMP levels increased. After delivery, both levels returned to pre-pregnancy levels. The authors hypothesize that enzymes are likely to be the cause of this shift, but further research needs to confirm this suggestion. Especially thiopurine S-methyltransferase and NUDT15 play a key role in this metabolism. Despite the increase of the 6-MMP levels, thiopurines are not considered teratogenic in humans [10,11]. Twelve studies covered the biologicals, in which data for five types of biologicals were presented. Except for one study, all studies reported only concentrations and no PK-parameter values. With the concentrations, the influence of pregnancy on the drug levels could be determined. The IFX levels increased, ADL levels stayed relatively stable and VDZ levels decreased during pregnancy. After pregnancy, the drug levels of the biologicals were lower compared to the pre-pregnancy levels. The IFX levels in Figure 4 showed discrepancies at T5 for two studies [13,18]. The discrepancies at T5 may possibly be related to the time of measurement after delivery. The latest measurement performed by the outliers, Kane et al. and Vasilauskas et al., was at 14 weeks [13,18]. Seow et al. and Flanagan et al. defined post-pregnancy as up to 6 months [14,15]. Grisic et al. showed measurements up to 250 weeks after conception, and Steenholdt et al. made their last measurement at 28 weeks after delivery [17,21]. For UST and GLM, the data were too scarce to observe a trend in concentration during the pregnancy. The reasons for these trends remain unclear. Some suggestions were made about the size of monoclonal antibodies. Due to their high molecular size and hydrophilic characteristics, the biologicals tend to have a small volume of distribution, limited to the plasma and extracellular fluids. One could argue that the increased plasma volume in a pregnant woman has an impact on the PK of the monoclonal antibodies, but since the volume of distribution for biologicals is small, consequentially the PK of monoclonal antibodies is not altered. [15]. For corticosteroids, no studies were found for pregnant women with IBD. However, from the literature search, five articles concerning betamethasone (BET) and two articles covering prednisone and prednisolone were found for other indications than IBD [28,29,30,31,32,33,34]. However, in the clinical setting, sometimes dosing advice needs to be determined for drugs (e.g., corticosteroids) that are not investigated in pregnant women with IBD. Investigation of the PK of a drug used in a population of pregnant women through an alternative route or for a different indication can be a helpful first step. The article characteristics are available in Appendix C Table A3, and the results are available in Appendix C Table A4. In contrast to the thiopurine studies and studies with biologicals, the corticosteroid studies for other diseases than IBD presented their data in PK parameters instead of concentrations. The same trend in dose-independent PK changes in corticosteroids during pregnancy was found in several studies [32]. All studies mentioned that the clearance of BET increased during pregnancy. This is likely to be originating from the CYP3A4 enzyme and 11β-hydroxysteroid dehydrogenase 2 (11β-HSD2) activities [32,33,34]. Prednisone and prednisolone are affected by these enzymes, too.

When specifically focusing on the limitations of the included studies in this review, the studies with thiopurines and biologicals only presented sparse drug concentrations over time. These specific data give insight into the behavior of these drugs during pregnancy, but a more complete view of the PK parameters would be desirable. One of the reasons for the lack of PK parameters generated from the obtained concentrations over time for thiopurines, as well as biologicals, is the fact that often only one blood sample per patient per trimester has been determined. PK parameters cannot be calculated based on just one sample per trimester in such a small study population. Population PK modeling can be a useful tool, not only to predict PK parameters, but also to develop more evidence-based dosing in special populations, such as pregnant women and fetuses [35]. The advantage of such a population PK model is that all individual concentrations of all patients will be analyzed together in a population setting, while, at the same time, data from individual patients are still distinguishable. Inter- and intra-patient variability can still be characterized. The advantage of this technique is that no complete PK profile of thiopurines or biologicals per patient, for example, is needed. The patient-related (i.e., age, trimester, weight, disease state and single vs. twin pregnancy) and treatment characteristics (i.e., route of administration) can thereby be used to (partly) understand and explain the inter-individual and intra-individual variability in these pharmacokinetics parameters in pregnant women. Therefore, those covariates can be used to determine if and how dosing can be individualized. After the development of such a pharmacokinetics model, the dosage needed to reach a specific target concentration can be developed. After the development of a PK model and model-based dosing, it would be of the utmost importance to prospectively validate the model-based dosing in a clinical study, not only to investigate whether the target concentration is reached, but also to investigate if the safety values are within the reference range. A first step could be to evaluate the already performed PK studies on quality and the amount of data, including clinical characteristics, drug concentrations in plasma, number of patients and time of sampling, retrieved from these studies in order to perform a pooled-PK analysis [35]. Such a pooled analysis has already been performed by Colin et al. for vancomycin in other special populations, with the aim to study all common covariates in adults in datasets on intravenous vancomycin [36]. In this way, a pooled analysis could be performed with all PK data of the pregnant population. After developing a PK model specific for pregnant women, a next step could be to design a new study with a specific focus on, for example, additional covariates that have not yet been studied in already published datasets and that could possibly explain the residual variability. In this way, we should use these already available datasets and published population PK models to put new datasets into these perspectives. This is an effective approach to explore additional covariates or specific subpopulations, but it should be preceded by a critical assessment of the published models [35].

Furthermore, in this review, the focus was on the PK of IBD drugs used by pregnant women with IBD. Therefore, the effect of the drugs on the fetus was not in the scope of this review. However, for IBD drugs transferring the placenta (e.g. thiopurines, ADL and IFX) fetal exposure as well as fetal outcomes (e.g., safety-related parameters) are important to monitor. Within the group of the thiopurines, Jharap et al. [11] reported that thiopurine exposure may cause neonatal anemia. This outcome, however, was not supported by Flanagan et al. [10]. Those authors reported that 80% of the infants at 6 weeks of age showed neonatal thrombocytosis and abnormal liver function [10]. When more data are gathered, a more conclusive statement can be made. In regard to the biologicals, these drugs are not linked to short-term severe adverse outcomes. On the other hand, these drugs are relatively new, and, therefore, the long-term outcomes are yet to be uncovered. Drugs such as IFX, which belong to the IgG1 subfamily, are actively transported over the placenta and, thus, expose the fetus to these drugs [23]. The corticosteroids, although mostly investigated in pregnant women with another indication than IBD, did not show any life-threatening adverse outcomes. The fetus seems to be protected by the more prevalent 11β-HSD2 enzyme, which turns the active prednisolone into inactive prednisone. Compared to the maternal body, in which the 11β-HSD1 enzyme is more present, the opposite drug ratio is observed [30,31]. The ECCO states that some risks, such as orofacial malformations, are found in the newborns, but with a small risk. Despite the low risk of serious adverse outcomes for both the newborn and mother, clinicians should be aware of the potential risks that corticosteroids could cause. Due to the potential risks, the use of corticosteroids is reserved only for case of flares.

## 5. Conclusions

In conclusion, we conducted a systematic review of the literature containing the available values of PK parameters related to IBD-drugs in pregnant women with IBD. We found relevant studies that presented the results for aminosalicylates, thiopurines and biologicals. In general, no PK values were found other than concentrations. Thiopurine metabolite concentrations tend to alter per consecutive trimester, while biologicals show that the concentrations are either rising, remain stable or are decreasing depending on the specific biological. Studies concerning corticosteroids presented values for a wide variety of PK parameters, but they did not include IBD pregnant women. We confirmed that there is a knowledge gap concerning the PK of IBD-related drugs in pregnant women with IBD. In the future, more PK studies on IBD drugs have to be performed in order to develop evidence-based dosing.

## Figures and Tables

**Figure 1 pharmaceutics-14-01241-f001:**
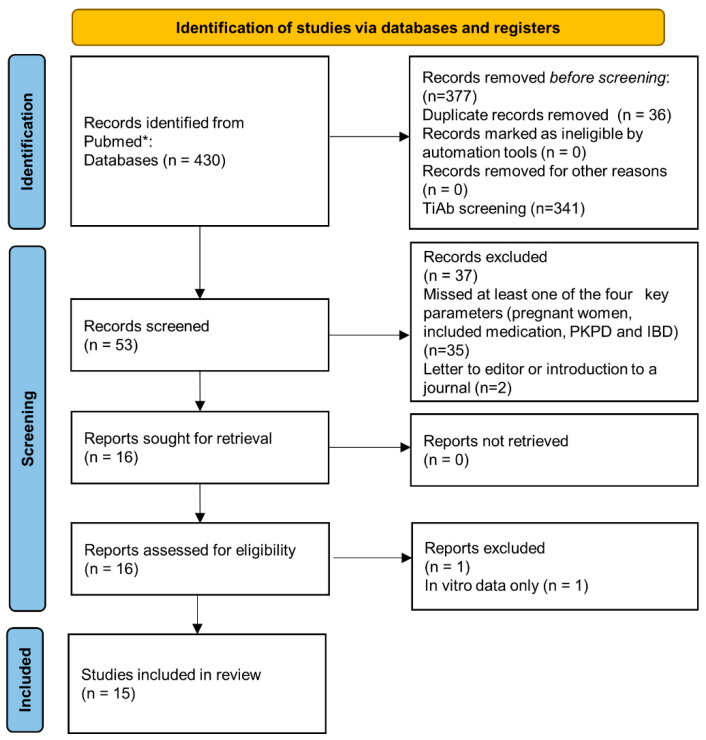
PRISMA flow diagram [8].

**Figure 2 pharmaceutics-14-01241-f002:**
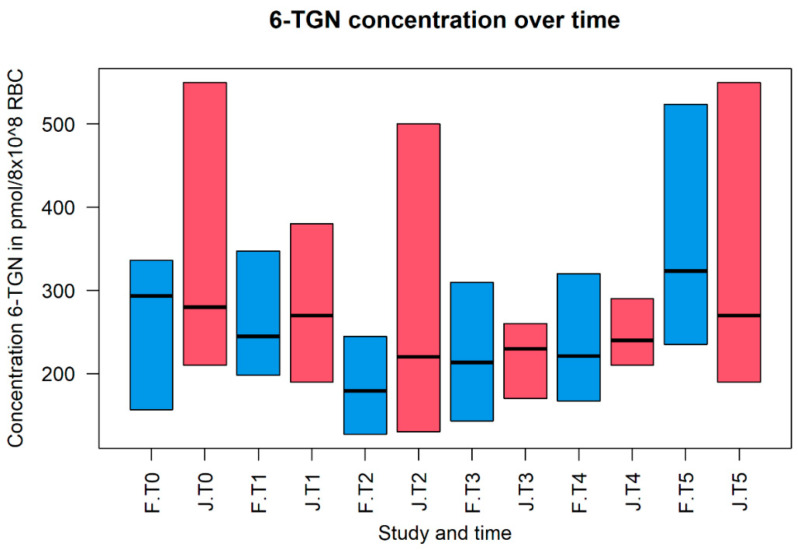
The concentration of 6-thioguanine nucleotides (6-TGN) during the different states of pregnancy. Concentrations of 6-TGN are expressed in pmol × 10^8^ Red Blood Cell (RBC) count on the *y*-axis (median with corresponding 25th and 75th percentile). The different states of pregnancy per study are shown on the *x*-axis. The different states are expressed as pre-pregnancy (T0), trimesters one until three (T1, T2 and T3), delivery (T4) and postnatal (T5). F, the blue bar, represents the study of Flanagan et al. (2021); and J, the red bar, represents the study of Jharap et al. (2013) [10,11].

**Figure 3 pharmaceutics-14-01241-f003:**
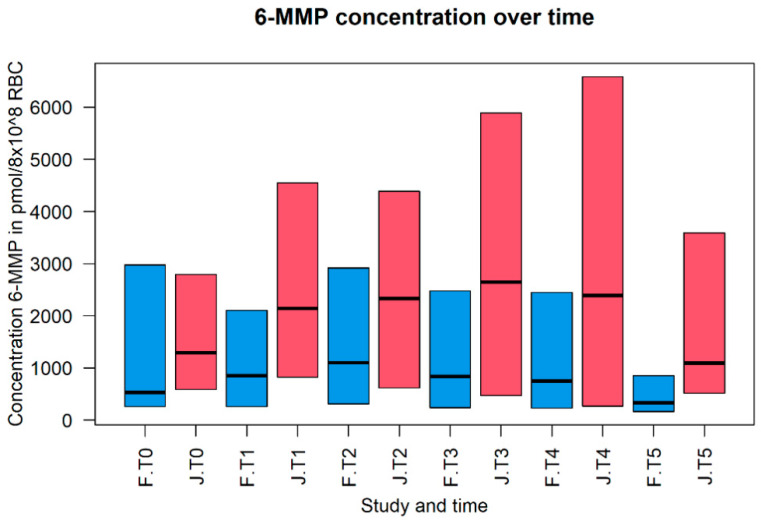
The concentration of 6-methylmercaptopurine (6-MMP) during the different states of pregnancy. Concentrations of 6-MMP are expressed in pmol × 10^8^ Red Blood Cell (RBC) count on the *y*-axis (median with corresponding 25th and 75th percentile). The different states of pregnancy per study are shown on the *x*-axis. The different states are expressed as pre-pregnancy (T0), trimesters one until three (T1, T2 and T3), delivery (T4) and postnatal (T5). F, the blue bar, represents the study of Flanagan et al. (2021) and J, the red bar, represents the study of Jharap et al. (2013) [10,11].

**Figure 4 pharmaceutics-14-01241-f004:**
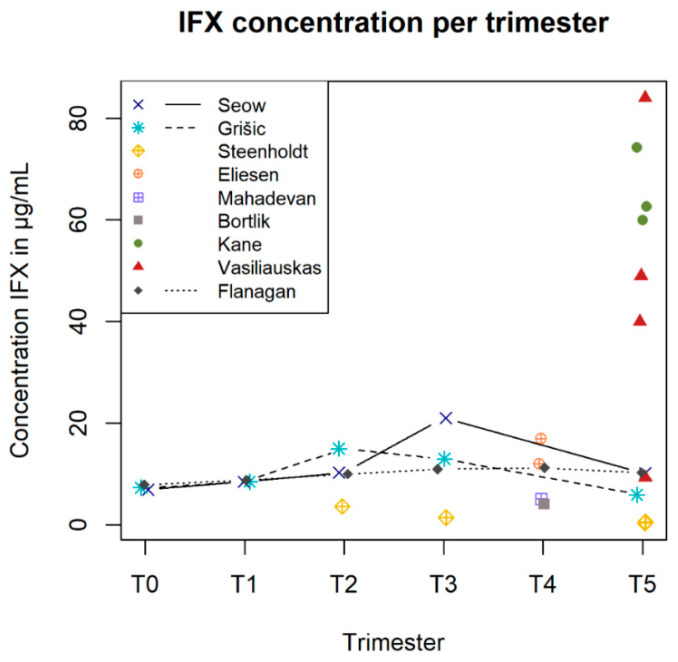
The concentrations of infliximab (IFX) from all available studies [13,14,15,16,17,18,19,20,21] in μg/mL (shown on *y*-axis) during the different stages of pregnancy (shown on *x*-axis). The different states are expressed as pre-pregnancy (T0), trimesters one until three (T1, T2 and T3), delivery (T4) and postnatal (T5).

**Figure 5 pharmaceutics-14-01241-f005:**
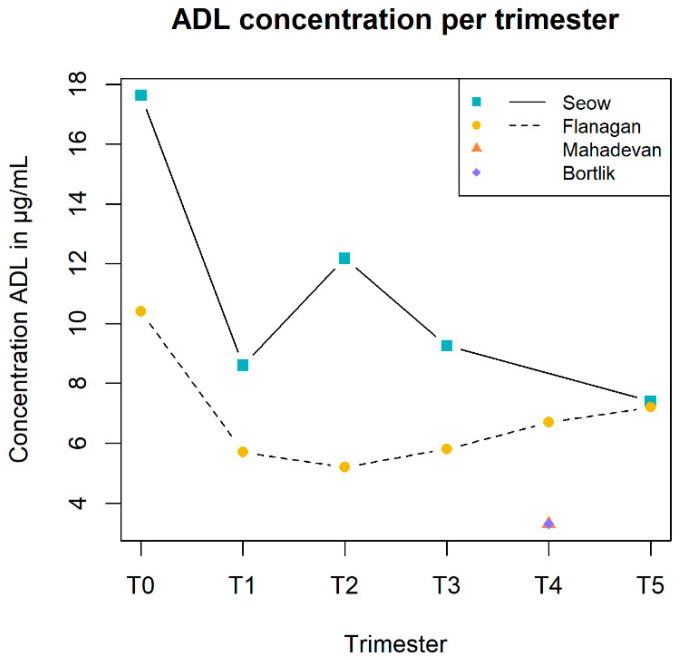
The concentrations of adalimumab (ADL) from all available studies [14,15,19,20] in μg/mL (on the *y*-axis) during the different stages of pregnancy (on the *x*-axis). The different states are expressed as pre-pregnancy (T0), trimesters one until three (T1, T2 and T3), delivery (T4) and postnatal (T5).

**Figure 6 pharmaceutics-14-01241-f006:**
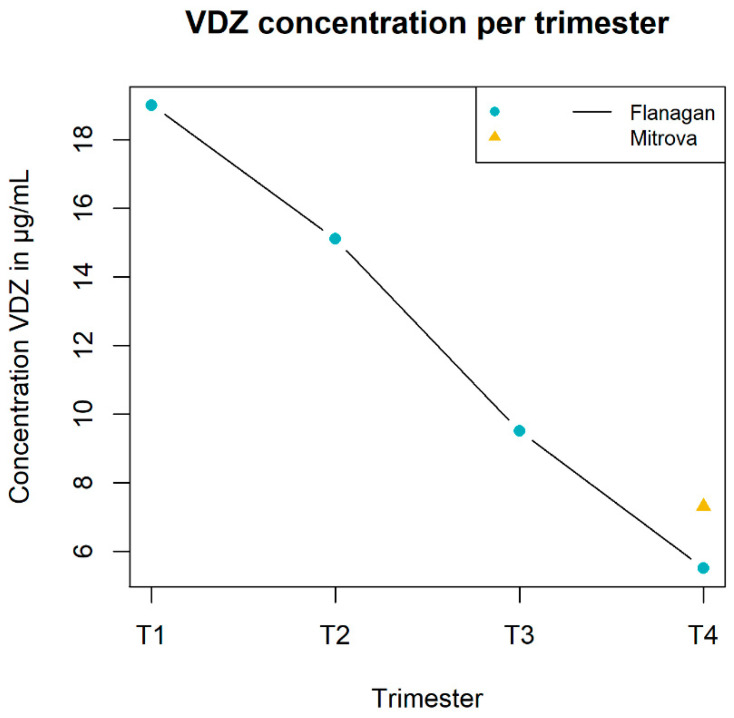
The concentrations of vedolizumab (VDZ) from all available studies [15,24] in μg/mL (on the *y*-axis) during the different stages of pregnancy (on the *x*-axis). The different states are expressed as pre-pregnancy (T0), trimesters one until three (T1, T2 and T3) and delivery (T4).

**Table 1 pharmaceutics-14-01241-t001:** Key terms with corresponding field used in the search strategy.

Pharmacokinetics	IBD-Related Medication	Pregnant Women
“Pharmacokinetics” [Mesh] Pharmacokinetic * [tiab], “drug kinetic *” [tiab], ADME * [tiab], LADMER [tiab], absorption [tiab], distribution [tiab]. Metabolism [tiab], elimination [tiab],	*“Exact drug name” [Mesh]* For further specifications per drug, see Appendix A Table A1.	“Pregnancy” [Mesh] Pregnanc * [tiab], gestation * [tiab], caesarean * [tiab], cesarean * [tiab], “abdominal deliver *” [tiab], “C-section *” [tiab], “Delivery, Obstetric” [Mesh], “obstetric deliver *” [tiab], “Labor, Obstetric” [Mesh], “obstetric labor” [tiab], labor [tiab], labour [tiab]

**Table 2 pharmaceutics-14-01241-t002:** Summary of the pregnancy-induced changes in the pharmacokinetics of IBD-related drugs. The data are presented as mean (SD), median (IQR) or alternative method, indicated next to the corresponding value. Each row is dedicated to a medicine. When a column overlaps multiple rows, the data are shared over multiple rows.

Author/s (Year) [Reference]	Medication	Ctrough	C Unspecified	Study Conclusion	Dose Advice	Remarks
**Aminosalicylates**						
Christensen et al. (1993) [9]	Mesalazine Pentasa suppository	-	μmol/L 0.08	-	-	The interval between the last intake of the drugs and the delivery was between 5 and 24 h. The data were extracted via a plot digitizer.
Christensen et al. (1993) [9]	Mesalazine Mesasal tablet and suppository	-	μmol/L 1.42			
Christensen et al. (1993) [9]	Mesalazine Pentasa tablet	-	μmol/L Patient 1: 2.6 Patient 2: 0.5 Patient 3: 0.2			
**Thiopurines**						
Flanagan et al. (2021) [10]	AZA	-	6-TGN pmol/8 × 10^8^ RBCs T0 = 293.5 (156.5–336.5); 16 obs T1 = 245.0 (198.0–347.5); 24 obs T2 = 179.0 (127.0–245.0); 35 obs T3 = 213.5 (143.0–310.0); 30 obs T4 = 221.0 (167.0–320.0); 25 obs T5 = 323.5 (235.0–524.0); 30 obs 6-MMP pmol/8 × 10^8^ RBCs T0 = 529.0 (258.0–2974.5); 16 obs T1 = 851.0 (255.5–2104.0); 24 obs T2 = 1103.0 (312.0–2919.0); 35 obs T3 = 838.0 (236.0–2474.0); 30 obs T4 = 747.0 (228.0–2451.0); 25 obs T5 = 329.5 (160.0–854.0); 30 obs	The 6-TGN median levels in T2 were significantly lower than observed from T0 to T5 (*p* < 0.001). This was still the case when adjusted for patient weight during pregnancy. The median 6-MMP levels increased significantly in T2, looking at T0 to T5 (*p* < 0.01).	When considering an increase in thiopurine dosing during pregnancy, extra attention should be paid to TDM, since this study observed an increase in 6-MMP levels with even the slightest change in dose elevation.	Data were included only if at least two observations between T0 and T5 were available; on stable dosing, for at least four weeks before testing. Two patients were excluded due to a dose change between T0 and T5. The total amount of participants included in the study went from 42 to 40.
Flanagan et al. (2021) [10]	MP	-				
Jharap et al. (2013) [11]	AZA	-	6-TGN pmol/8 × 10^8^ RBCs T0 = 280 (210–550) T1 = 270 (190–380) and 220 (130–500) T3 = 230 (170–260) T4 = 240 (210–290) T5 = 270 (190–550) 6-MMP pmol/8 × 10^8^ RBCs T0 = 1290 (584–2790) T1 = 2140 (820–4548) and 2330 (615–4390) T3 = 2648 (468–5888) T4 = 2390 (268–6588) T5 = 1090 (518–3590)	Over the whole pregnancy, median 6-TGN levels were decreasing significantly (*p* = 0.001). After delivery, the 6-TGN levels normalized to pre-pregnancy levels. The 6-TGN levels in T1.2 were significant lower compared to T0 (*p* < 0.05).	-	-
Jharap et al. (2013) [11]	MP	-				
**Biologicals**						
Kane et al. (2009) [13]	IFX	-	μg/mL T5: Patient 1: 74.27 Patient 2: 62.62 Patient 3: 59.97	-	-	Time between infusion and measurement - Patient 1: 6 days Patient 2: 5 days Patient 3: 43 days
Seow et al. (2017) [14]	IFX	μg/mL T0: 6.9 T1: 8.5 (7.23–10.07); 5 obs T2: 10.31 (7.66–15.63); 15 obs T3: 21.02 (16.01–26.70); 16 obs T5: 10.17 (6.80–15.50)	-	Intra-partum, albumin levels decreased (*p* < 0.05), BMI increased (*p* < 0.05) and CRP stayed stable (*p* > 0.05). There was an inverse relationship between infliximab levels and CRP in CD (*p* = 0.03). After adjusting for albumin, BMI and CRP, gestational age had a significant effect on the IFX concentrations with multivariate mixed modeling (*p* = 0.02).	The authors suggest that anti-TNF levels can be targeted to the lower end of the therapeutic range during T0 in clinical stable patients. The levels should be monitored again at T2 to inform the clinician if a third dose is necessary in T3. The regimen used in T0 should be continued in T5.	T0 and T5 of IFX and all T values of ADL are extracted via a plot digitizer.
Seow et al. (2017) [14]	ADL	μg/mL T0: 17.63 (16.01–19.98) T1: 8.6 (0–15.65) T2: 12.18 (7.72–16.95) T3: 9.26 (0.79–12.84) T5: 7.40 (1.66–13.70)	-	Intra-partum, albumin levels decreased (*p* < 0.05), BMI increased (*p* < 0.05) and CRP stayed stable (*p* > 0.05).		
Flanagan et al. (2020) [15]	IFX	μg/mL T0: 7.9 (6.3–11.0); obs 6 T1: 8.8 (5.5–12.4); obs 15 T2: 10.0 (7.1–13.7); obs 30 T3: 11.0 (7.1–16.8); obs 20 T4: 11.2 (8.4–15.7); obs 8 T5: 10.3 (4.3–13.8); obs 12	-	The median albumin level from T1 to T3, respectively, 36.0, 30.5 and 28.0 g/L, decreased significantly (*p* < 0.001). A small significant increase in IFX levels per gestational week of 0.16 (95% CI 0.08 to 0.24) μg/mL was observed (*p* < 0.001).	Routine TDM and dose adjustments are not recommended because the predicted alterations in concentration were small. Therefore, the change in concentrations was unlikely to be clinically relevant.	-
Flanagan et al. (2020) [15]	ADL	μg/mL T0: 10.4 (10.0–10.8); obs 2 T1: 5.7 (4.8–10.2); obs 9 T2: 5.2 (4.0–6.8); obs 12 T3: 5.8 (4.8–8.0); obs 14 T4: 6.7 (5.1–8.0); obs 8 T5: 7.2 (4.3–9.7); obs 8	-	The median albumin level from T1 to T3, respectively, 33.5, 30.0 and 27.0 g/L, decreased significantly (*p* < 0.001).		-
Flanagan et al. (2020) [15]	VDZ	μg/mL T1: 19.0 (13.0–23.0); obs 5 T2: 15.1 (8.6–21.7); obs 16 T3: 9.5 (3.7–20.0); obs 9 T4 5.5 (1.1–9.9); obs 2	-	A small significant decrease in VDZ levels per gestational week of −0.18 (95% CI −0.33 to −0.02) μg/mL was observed (*p* = 0.03).		From the 17 patients at the start, 12 were included.
Eliesen et al. (2020) [16]	IFX	-	μg/mL T4: patient 1, T4: 12.0 patient 2, T4: 17.0	-	-	Time between last dose to delivery for patient 1 was 57 and for patient 2 31 days
Steenholdt et al. (2011) [17]	IFX	μg/mL T2: 3.6 T3: 1.4 T5: 0.6 and 0.34	-	-	The authors found that an IFX concentration of 0.5 μg/mL and higher is associated with maintained response in both CD and UC. They suggest this as a valid cut-off level for clinically relevant IFX concentrations.	Infusions happened at 20 and 31 weeks of GA. After delivery, only two infusions are specified with the corresponding date. It should be noted, that at infusion 12, which is 4 infusions later than the last measured concentration, had a high Ctrough of 2.1 μg/mL T5, concentration of 0.6 μg/mL is extracted via a plot digitizer.
Vasiliauskas et al. (2006) [18]	IFX	-	μg/mL T5: 40 (week 2), 9.3 (week 10), 84 (week 13) and 49 (week 14)	-	-	Dosing happened at week 2 and 10 after delivery with infusions of 10 mg/kg IFX. The concentration from week 14 is obtained via a plot digitizer.
Mahadevan et al. (2013) [19]	IFX	-	μg/mL T4: 5.1 (3.8–16.5)	-	It should be taken into consideration to avoid IFX and ADL use 4 to 8 weeks before delivery in order to keep the placental transfer rate as low as possible. This advice is only applicable if the mother is in stable remission.	The median time from the last dose to delivery was 35 (14–74) days
Mahadevan et al. (2013) [19]	ADL	-	μg/mL T4: 3.3 (2.2–6.05)	-		The median time from the last dose to delivery was 38.5 (7–42) days. The authors mention that CZP levels in a newborn are minimal and therefore could be a good alternative to IFX and ADL.
Bortlik et al. (2013) [20]	IFX	-	μg/mL T4: Mean 4.1 {range: 0.0–18.0}	-	The authors recommend ceasing the therapy in the end of T2 or early T3 to minimize the exposure of IFX and ADL to the child.	One patient intensified the dosing regimen from each 8 weeks to each 6 weeks from gestational weeks 18 to 30.
Bortlik et al. (2013) [20]	ADL	-	μg/mL T4: 0.8 (0.0-2.5)	-		-
Grišić et al. (2020) [21]	IFX	mg/mL/kg T0: 7.3 (2.0–11.6); obs 119 T1: 8.5 (1.4–11.5); obs 16 T2: 15.0 (9.8–20.5); obs 18 T3: 13.0 (6.5–35.8); obs 7 T5: 5.9 (3.3–11.1); obs 12	-	A significant increase in IFX maternal levels was shown in T2 compared to T0 (*p* = 0.003) and T1 (*p* = 0.04).	It is necessary to continue the IFX therapy in late T2 or early T3 to maintain a constant maternal IFX concentration until the end of the pregnancy, if desired. In the case of continuation of IFX therapy in the last part of the pregnancy, TDM can guide to a balanced and lower dose regime.	Concentrations are presented as dose-normalized Ctrough concentrations. Clearance was determined to be 0.608 L/d. Anti-IFX antibodies were accountable for an increase of 69% in clearance. Volume of distribution was determined to be 18.2 L.
Sako et al. (2021) [22]	UST	-	ng/mL T4: 267.7	-	-	The last dose UST was at week 23, day 3 GA.
Benoit et al. (2018) [23]	GLM	-	μg/mL T4: 6.6	-	-	Dose-delivery interval was three days.
Mitrova et al. (2021) [24]	VDZ	-	μg/mL T4: 7.3 (2.9–17.9)	A significant correlation for VDZ was found between maternal drug level and gestational week of the last administration (ρ = 0.751, *p* = 0.001). Another correlation for VDZ was found between maternal drug level and the interval between the last infusion and delivery (ρ = −0.917, *p* < 0.001).	-	The therapy was intensified by 1 individual.
Mitrova et al. (2021) [24]	UST	-	μg/mL T4: 5.3 (2.3–10.1)	Between maternal UST levels and gestational week of last administration, there was a significant correlation (ρ = 0.578, *p* = 0.02).	-	The therapy was intensified by 5 individuals.

**Abbreviations**: *6*-*TGN*, 6-thioguanine nucleotides; *6-MMP*, 6-methylmercaptopurine; *ADL*, adalimumab; *AZA*, azathioprine; *BMI*, body mass index; *CD*, Crohn’s disease; *CRP*, c-reactive protein; GA, gestational age; *GLM*, golimumab; *IBD*, inflammatory bowel disease; *IFX*, infliximab; μg/mL, microgram per milliliter; μmol/L micromol per liter; *MP*, mercaptopurine; ng/mL, nanogram per milliliter; *OBS*, observations; pmol/RBC, picomoles/red blood cells; *T0*, pre-pregnancy; *T1*, trimester 1; *T2*, trimester 2; *T3*, trimester 3; *T4*, during delivery; *T5*, postpartum; TDM, therapeutic drug monitoring; *UC*, ulcerative colitis; *UST*, ustekinumab; *VDZ*, vedolizumab.

## Data Availability

Not applicable.

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
