# Peer review of "The Effect of Pregnancy and Inflammatory Bowel Disease on the Pharmacokinetics of Drugs Related to Inflammatory Bowel Disease—A Systematic Literature Review"

_pharmaceutics, 2022, doi:10.3390/pharmaceutics14061241_

Round 1

Reviewer 1 Report

  • “Inflammatory Bowel Disease (IBD) is an overarching term for chronic inflammation in the gastrointestinal tract. IBD is characterized by exacerbations.”

Add a reference (for example: “Actis GC et al. History of Inflammatory Bowel Diseases. J Clin Med. 2019 Nov 14;8(11):1970.”)

  • Use oxford comma

  • What do you mean for “changes in genetics”?

  • “A consensus exists among clinicians to resume the treatment of pregnant women with IBD”

Resume?!

  • “A total of 430 studies were identified. After removal of duplicates (n=36), titles and abstracts of 53 studies were screened.”

Why 53 and not 394 (430 - 36)?

  • “Looking at the other studies presenting a dose advice for IFX, the following results were found. Four studies mentioned that dosing for IFX should be halted at the end of the second trimester or the beginning of the third trimester”

Why stop? Is this linked with mother trough levels?

  • Report the PK data about vedolizumab in the text

  • English must be reviewed (for example “this is the first time a systematic review”)

  • “Despite the increase of the 6-MMP levels, thiopurines are not considered teratogenic”

In humans or mice?

  • “In contrast to the thiopurine studies and studies with biologicals, the corticosteroid studies presented their data in PK-parameters instead of concentrations.”

In IBD or other diseases?

  • “However, as mother and fetus are inseparable from each other,”

It depends on the drug and on the trimester. For example, infliximab and adalimumab do not cross the placenta in the first two trimesters, certolizumab do not pass the placenta during the whole pregnancy.

Reviewer 2 Report

The authors provide comprehensive analysis of IBD medication used during pregnancy. The fllowing is my comment

1. Page 15  Table Flanagan dosing of VED is q1d ? The dose is unusual and please check

Reviewer 3 Report

I read the manuscript "The Effect of Pregnancy and Inflammatory Bowel Dis-ease on the Pharmacokinetics of Drugs Related to Inflammatory Bowel Disease- A Systematic Literature Review" with great interest.

It is a well-written, well-structured, and scientifically sound manuscript. I have a few minor comments:

- I recommend authors to give the conclusive summary at the end of every section/medication like thay did in the section 3.3. thiropurines starting from line 204. They dont have the same for section 3.2. for aminosalicylates for example. There they only summarise the one study they found but that is all, I would recommend that they add a conclusive sentence about the existing evidience at the end of every section. If it is not possible to have a conclusion then that is the conclusion itself. 

- I kindly believe that although Table 2 is necessary;  it is way too large for the publication. It is hard to read and distracts the reader. I recommend authors to have this table as supplementary file, by doing this they help readers to stay on the focus while they read.

- I recommend author to discuss their methodological limittions in addition to the limitations of the studies that included. For example thy only seached for one database, this is a limitation if one looking for all the published evidence. I normally would recommend to add more databases but then it might be a lot of work at this point so mentioning this as a limitation is necessary. 
